# Mechanical Properties of 3D Printed Orthodontic Retainers

**DOI:** 10.3390/ijerph19095775

**Published:** 2022-05-09

**Authors:** Marcel Firlej, Katarzyna Zaborowicz, Maciej Zaborowicz, Ewa Firlej, Ivo Domagała, Daniel Pieniak, Joanna Igielska-Kalwat, Artur Dmowski, Barbara Biedziak

**Affiliations:** 1Department of Orthodontics and Craniofacial Anomalies, Poznań University of Medical Sciences, Collegium Maius, Fredry 10, 61-701 Poznan, Poland; marcel-firlej@wp.pl (M.F.); efirlej@ump.edu.pl (E.F.); ivo.m.domagala@gmail.com (I.D.); joanna.igielska@wp.pl (J.I.-K.); biedziak@ump.edu.pl (B.B.); 2Department of Biosystems Engineering, Poznan University of Life Sciences, Wojska Polskiego 50, 60-627 Poznan, Poland; 3Faculty of Transport and Computer Science, University of Economics and Innovations in Lublin, Projektowa 4, 20-209 Lublin, Poland; daniel.pieniak@wsei.lublin.pl (D.P.); artur.dmowski@wsei.lublin.pl (A.D.)

**Keywords:** digital orthodontics, lingual retainer, 3D printing in orthodontics, CAD/CAM in orthodontics, aligner

## Abstract

Orthodontic retention is the final important stage of orthodontic treatment, the aim of which is to consolidate the functional and aesthetic position of teeth. Among adults, fixed retainers made of different types of wires are the most common. The aim of this study was to analyse the mechanical properties of a new generation of fixed orthodontic retainers—printed by 3D printers. Materials and Methods: The study was conducted using samples made of Nextdent MFH C&B N1 resin in the form of cuboid bars with nominal dimensions of width b = 3 mm, thickness d = 0.8 mm; 1 mm; 1.2 mm, length l = 30 mm for each type. The influence of the thickness of the retainers on their strength under loaded conditions was evaluated. Flexural strength, elastic properties, deflection, and creep were compared. The samples were aged in an artificial saliva bath at 37 ± 1 °C during the strength tests. Results: It was shown that differences in the thickness of the samples affected their elastic and strength properties. The highest average flexural modulus, the highest deflection, creep, and strength was characteristic of the samples with the highest thickness (1.2 mm). Samples with an average thickness of 1 mm had the lowest modulus of elasticity. Conclusions: The mechanical properties of 3D printed retainers show that they can be an alternative to metal retainers and the procedure of making new retainers, especially when patients have aesthetic requirements or allergies to metals.

## 1. Introduction

Retention type planning is done individually for the patient at the beginning of orthodontic treatment. After analysis of the malocclusion, determination of the clinical problems, type of biomechanics and developmental age of the patient, the retention treatment protocol suitable for the patient can be planned, including the time and form of retention braces. Clinicians’ opinions on the necessity of a retention phase in orthodontic treatment have changed over the years and are the subject of scientific disputes. In recent years, it has been emphasised that retention is an important factor in the prevention of post-treatment orthodontic recurrence.

The retention period varies between all patients. Some patients are recommended for life-long retention. Among growing patients, orthodontists more often use removable than fixed retention appliances. They may be in the form of removable appliances with metal arches on the labial and buccal surfaces of the teeth, such as the Hawley appliance, or in the form of transparent thermoformable appliances covering all tooth surfaces, including the alveolar process. Those transparent appliances are especially well accepted by patients because of their cosmetic value. Hawley’s plates can be made using a variety of technologies, including compression moulding, 3D printing and thermoforming. The most significant differences in mechanical properties are observed in thermoformed devices [1,2,3]. Fixed retention appliances are mainly used among adult patients. They are also used among growing patients who do not wear removable retention for the required time.

Originally, round or rectangular wires were used as stabilisers. They were shaped on models or cast in the laboratory. Later, they were replaced by multi-stranded braided wires, which are more flexible and allow physiological tooth mobility. The elimination of metal presence in the oral cavity, especially those remaining in it for many years, has opened the way for researchers to search for alternative materials that could be used in fixed retention appliances. There has been introduced fibreglass bonded to each tooth, using composite techniques, with acid etching of the enamel [4]. Although they were aesthetically acceptable and non-metallic, they showed a higher failure indicator in maintaining bonding to enamel, and their rigidity prevented physiological tooth mobility. This has resulted in a significant reduction in their use in permanent orthodontic retention. The most common use of retainers is on the lower incisors, as these are the most likely to relapse. A 20-year evaluation after treatment showed that during the first 10 years, the recurrence rate is higher than during the second 10 years [5]. Different shapes and sizes of anterior retainers can result in different degrees of plaque accumulation. Modified multi-loop fixed retainers cause more periodontal changes than simple arch shapes [6]. The use of prefabricated universal fibreglass retainers causes a higher risk of failure due to the fact that it is a technique that depends on the operator’s skills [7]. There are other methods to establish orthodontic treatment stability. A lot of attention has been given to low-intensity pulsed ultrasound, which, by accelerating the healing of bone tissue, consolidates the treatment effect more quickly, making retention shorter [8].

In recent years, the development of new technologies, especially the development of CAD-CAM systems, has resulted in the development of new materials used in medicine and orthodontics. The use of this technology also makes it possible to create permanent nickel-titanium retainers [9]. Zachrisson, in 2018, presented a new type of fixed retainer made with digital technology using a milling method from PEEK material [10]. A Beretta publication describes the full process of constructing and seating such a restoration [11]. Polyetheretherketone (PEEK) has a high tensile strength of 90 to 100 MPa with an elastic modulus of 3.6 MPa [12]. Retainers made of PEEK can be produced by CAD/CAM milling or printing [13]. Retainers can also be made of resins using 3D printing technology. As this is a brand new technology, the study of the properties of devices made from these materials is not well known. Depending on the resins used for 3D printing in dentistry, their properties may vary [14]. The use of this type of retainer may be an option for patients with metal allergies or those with high aesthetic expectations as an alternative to fibre-reinforced composite (FRC) retainers. An additional advantage of this type of solution is the absence of metal elements that could influence the magnetic resonance image [15]. During designing such a retainer, the missing tooth can be created at the same time as the retainer. This is particularly important when the young patient has to wait for the implant to complete its development or the treatment requires longer stabilisation. Previously, this type of solution was based on the method of cementing tooth spans to fibreglass splints [16]. Due to the mechanism of adhesion, different bonding techniques can be used for both spot- and full-bonded 3D printed retainers like in the case of FRC [17]. Due to the individual shape of the retainers, the amount of adhesive can be significantly less than recommended, e.g., in multi-stranded wires, the recommended thickness of the composite adhesive is between 2–4 mm [18].

The new retainer, designed digitally and created in 3D printing by individual design, is perfectly adapted to the patient’s teeth anatomy. The use of an intraoral scanner additionally eliminates the need to make impressions, take wax-ups, disinfect and prepare models. Each of these steps may cause inaccuracies in the following procedure but may also contribute to cross-infection [19]. 3D printers are already used to produce retainers. The most common way to use this technology is to scan the patient’s teeth, print out a model of the teeth and conventional vacuum-forming of the retainer. Using 3D-Printed Wearable Personalized Orthodontic Retainers for Drug release, for example, Clonidine Hydrochloride to treat high blood pressure is another important field of study [20].

A new approach that has been studied is the CAD/CAM design of printed individual, fixed retainers. Different dental resins used in the oral cavity also have different results in Shore hardness tests and tribological properties—scratch resistance and sliding wear resistance in corrosive conditions [21]. This is a fast and precise method, which enables the restoration to be designed in a few minutes using free software such as Meshmixer (Figure 1).

The aim of this study was to analyse the mechanical properties of a new generation of 3D printed fixed orthodontic retainers and to measure if the difference between sample thickness and their properties is other than zero.

## 2. Materials and Methods

Samples which were used were made in the form of rectangular beams with nominal dimensions of width b = 3 mm, thickness d = 0.8 mm; 1 mm; 1.2 mm, and length l = 30 mm, for each type. The size of the samples was chosen to match anatomical needs, i.e., tooth size. Samples were printed on a Phrozen MINI4k printer, using Nextdent MFH Crown and Bridge N1 resin. This resin is commonly used for printing restorations like crowns and bridges, and it has good mechanical properties at loading. According to a study, it has good mechanical properties after ageing in saliva in the oral cavity at 36.6 °C [14]. The influence of the thickness of the retainer on their strength under loading conditions was evaluated. Flexural strength, elastic properties, deflection and creep, were compared.

### 2.1. Flexural Strength and Flexural Modulus

In this study, the flexural strength test consisted of a three-point bending test, which is model-based [22]. This means that, in order to recognise the influence of material and thickness, simplifying conditions were adopted in comparison to real dental structures. For the study, there were made rectangular samples reinforced with long fibres with different weave architecture, according to the scheme presented in Figure 2, Table 1 and Table 2. The test was performed according to the codified method according to the technical standard ISO 10477. The samples used in the test were made in the shape of rectangular beams with nominal dimensions width b = 3 mm, thickness d = 0.8 mm; 1 mm; 1.2 mm, and length l = 30 mm. The thickness (height) and width of the samples were measured using a dial calliper. The samples were aged in a bath of artificial saliva during the strength test. The composition of the artificial saliva was based on the standard PN-EN ISO 10271: 2012. The temperature of the artificial saliva during the tests was 37 ± 1 °C, which was also due to the capacity of the heating system of the measuring vessel (Figure 1). The strength test was carried out using a Zwick/Roell Z100 universal testing machine; the crosshead speed was 1 mm/min, and the distance between supports was L = 20 mm. Figure 3 shows the scheme of the research system used in the bending test.

The strength (*σ*) was calculated from the following formula:(1)$$\sigma =\frac{3PL}{2bd^{2}}\;$$
where *P* is load during the test [N]; *L* is support span (mm); *b* is sample width (mm); *d* is sample thickness (mm).

### 2.2. Creep Test

Retainers are loaded for long periods of time. In such conditions, the properties of the polymeric material are different from those under short-term or one-time loading. This loading condition leads to deformation at lower stresses than that obtained in short-term strength tests [23,24]. The phenomenon of slow deformation of the material of an element under long-term, constant loads is called creep [25,26,27]. The creep tests were carried out according to the method specified in the ISO 899-2: 2005 technical standard titled “Plastics-Determination of creep characteristics-Part 2: Creep when bending under a three-point load” [28]. The creep tests were conducted in a medium reflecting real physiological conditions, i.e., in artificial saliva. The set-up for the creep test was the same as for the bending strength test (Figure 3).

The modulus of elasticity characterising the material’s ability of elastic—unstable deformations was calculated from the following formula:(2)$$E_{Y}=\left(\frac{P}{y}\right)\left(\frac{L^{3}}{4bd^{2}}\right)\;$$
where *P* is load during the test (N); *L* is spacing of supports (mm); *b* is sample width (mm); *d* is sample thickness (mm); *y* is beam deflection (mm).

The creep test termination criteria were also adopted. The test was terminated by the failure of the sample when the force dropped by 30% from the maximum or when the time limit was reached (30 min) if no other termination criterion was reached earlier. The creep modulus was measured at the specified test time intervals, i.e., 1, 3, 6, 12, and 30 min, at a load of 30 MPa. The creep modulus at the end of loading at the time intervals was calculated as follows:(3)$$E_{p}=\left(\frac{P}{y}\right)\left(\frac{L^{3}\cdot P}{4bd^{3}y}\right)\;$$
where *P* is load during the test at the end of the time interval (N); *L* is spacing of supports (mm); *b* is sample width (mm); *d* is sample thickness (mm); *y* is beam deflection at the end of the time interval (mm).

## 3. Results

### 3.1. Flexurar Strength and Elastic Module

Figure 4 presents the stress-deflection curves (sample deflection) from the three-point bending test. Stress is expressed in megapascals (MPa), while deflection is expressed as a percentage (%).

Table 3 shows the results of the three-point bending test put on retainer samples. The following values were obtained: *n*-size of the tested group, *E_f_*—modulus of elasticity, *σ*_0.2_—stress, with a sample deflection equal to 0.2%, *σ_fY_*—yield strength, *ε_fY_*—deflection corresponding to *σ_fY_*, *σ_fM_*—bending strength, *ε_fM_*—deflection corresponding to *σ_fM,_ σ_fB_*—stress at the time of destruction of the sample, *ε_fB_*—deflection corresponding to *σ_fB_*, *W_fM_*—work to *ε_fM_*, *W_fB_*—work to *ε_fB_*, $\bar{x}$—average, s—standard deviation, and ν—coefficient of variation.

### 3.2. Creep Test Results

Figure 5 and Figure 6 shows the results of the creep test. These are graphs of deflection of the samples in millimetres versus test time in logarithmic terms. Figure 4 shows the creep modulus of the samples in MPa as a function of the test time in logarithmic terms.

Table 4 shows the results of the creep tests. The following values were obtained: *E_t_*—flexural modulus of elasticity at bending, *σ**_t_*—stress load of the sample expressed in stress, *ε**_t_*—deflection of the sample under load *σ**_t_*, *σ**_fract_*—breaking stress, *ε**_fract_*—deflection of the sample at the time of destruction, and *τ**_fract_*—time to destruction.

## 4. Discussion

### 4.1. Flexurar Strength and Elastic Module Analysis

A in vitro study was presented. In the study, samples were made of material intended for retainers printed in 3D technology. This is a layer-by-layer printing technology in which layers are cured by photopolymerisation, meaning that energy is supplied to the process by a light beam [29]. This technology is effective due to its relatively high resolution, reasonable cost of 3D printers and materials and relatively high printing speed [30]. In the presented study, the thickness effect of samples printed using this technique was evaluated. Mechanical properties under bending load were compared. Using a three-point bending test, samples were measured under stress. It was shown that differences in the thickness of the samples affected their elastic and strength properties (Table 3 and Figure 4). The highest average bending modulus of elasticity was found in the specimens with the highest thickness (1.2 mm). But the lowest modulus of elasticity was characterised by specimens not with the lowest thickness but with a thickness of 1 mm. This mechanical property under bending load indicates that the elasticity is not directly proportional to the thickness of the samples. The modulus of elasticity is an important parameter for retainers, especially in the initial phases of retention. According to a study [31], immediately after completing orthodontic treatment, incisors show the greatest mobility and canines the lowest. Their mobility in the low-elasticity retainers could cause fractures of the bonded structures, especially in the initial phases of retention. Another important parameter was the bending strength. For this quantity, the ranking was similar to the elastic modulus. This means that the bending strength of 1 mm thick samples was more than twice as low as the bending strength of 1.2 mm thick samples. Moreover, it was several times lower than the bending strength of 0.8 mm thick samples, which was surprising. Thanks to the individual design of retainers using CAD/CAM technology, which is adapted to the shape of the teeth, their bending can be further minimised compared to standard retainers, which are adapted directly in the mouth. Another important parameter that is considered is work to failure (WfB). This value is considered by researchers to be one of the characteristics of prosthetic components, which are designed to absorb external loads [32], not only of physiological origin but also of special loads, with higher forcing energy values, e.g., associated with accidental mechanical trauma (accidental impact on an obstacle or a tool) [33]. The knowledge of the dissipative capacities—the main factors responsible for the development of damage, allows better use of the strength of the materials forming the load-bearing structure [34]. The high value of work to failure also translates into a non-catastrophic failure mechanism. A catastrophic mechanism was described in the paper [35]. Catastrophic damage to polymers is caused by voids and the initiation and propagation of brittle cracks [35]. Mechanically loaded elements, which have a high damage tolerance, do not show catastrophic damage, but the failure rate should be progressive [34], which can be verified on the basis of stress-strain diagrams (σ-which are shown in Figure 4). In these diagrams, it can be observed that the course of the curves has an interval shape close to monotonic, with the characteristics of a unimodal function. When the maximum stress is reached, which corresponds to the bending strength, there is no immediate failure. It is reached after a deflection by several additional percent. The highest deflection characterised the 1.2 mm thick samples (Figure 3, Table 3). The elastoplastic properties of the tested material are also responsible for this bending load behaviour. Ductile polymers fail by crazing or matrix shear yielding. Both mechanisms lead to high crack initiation energy [35]. This behaviour under load can also be explained by a relatively high capacity for irreversible deformation [36].

A characteristic feature of resin-based materials with extras is their possibility to become more fragile in time. The paper says that long-term clinical in vivo tests are crucial for verifying the outcome of wear resistance. It is essential that clinical studies should be preceded by long, simulated in vitro investigation in laboratory conditions. Moreover, loads and oral environmental factors should be included. Artificial saliva and in vitro study is one of the greatest limitations in this investigation, as it has been studied only in relation to the surface properties; ageing itself has been limited by the particular length of time. The authors want to improve their investigation by studying after a longer period of ageing and in different temperature conditions than a stationary one and in changing pH environment, which is more comparable to a real human oral cavity.

### 4.2. Creep Analysis

Creep is a result of the long time application of stress. This problem has critical importance for evaluating the durability and dimensional stability of medical polymer applications [37]. Tests under long-term load, which caused creep deformation, were performed in a bath of artificial saliva with a temperature of ~37 °C under load, which caused bending stress of 30 MPa. The long-term stress in the retainer is due to the adhesive bonding to the arch. This state of loading can lead to deformation and loss of rigidity of the retainer. The retainer material should therefore be able to withstand such loading and withstand simultaneous oral environmental factors. The susceptibility to creep is also dependent on the geometry of the component.

In the presented research, the thickness of the samples varied; the nominal length and width were the same. Under load, which corresponded to a stress of 30 MPa in the sample, the samples with the highest thickness (1.2 mm) had the lowest deformability. On the other hand, the highest deformability had the sample with a nominal thickness of 1 mm. Under a stress of 30 MPa, samples with a thickness of 0.8 mm were able to survive for about 3 min (Table 4), samples with a thickness of 1 mm for about 1 min, and samples with a thickness of 1.2 mm, survived the entire intended time range of the experiment—30 min.

Deformability is a quantity that is related to the rigidity of a material. Another measured quantity that determines the rigidity of a material is the flexural creep modulus (Et). The values of this quantity were the highest for samples with a thickness of 1.2 mm. The creep modulus decreased during the creep test. For 0.8 mm thick samples in stage 2 (Table 4), the modulus decreased by more than fifty percent. The 1 mm thick samples completely lost rigidity in stage 2 and failed. In contrast, the modulus of the 1.2 mm thick samples did not change significantly in stage 2 compared to the lower thickness samples. But in the final stage of the creep test (stage 5), the modulus dropped below one-third of the initial modulus.

Based on the research performed, it appears that the polymer resin-based material studied loses a large proportion of its properties under simulated loading and oral environment conditions. It was reported in [25] that at temperatures higher than room temperature and in a humid environment, creep increases. According to [38], the oral temperature range for men and women is 35.7–37.7 °C and 33.2–38.1 °C, respectively. According to another study [39], the oral temperature range is 36.3–37.1 °C among men and 36.5–37.3 °C among women. It corresponded to the experimental conditions. In a study [40], it was found that in physiological fluid environments, there may be a decrease in the creep resistance of polymers, which may be related to the diffusion of fluid particles between the polymer chains. These particles can, on the one hand, act as a plasticising agent and, on the other hand, as a stress corrosion inducer, which accelerates the material failure process [41]. The tested material retains its properties for a longer time, only in the case of 1.2 mm thick samples. However, considering the clinical application, it has to be assumed that long-term exposure to mechanical loads and oral environmental factors determines the effectiveness of a retainer made of the tested material. Furthermore, the damage in the form of deformation or loss of rigidity is not acceptable from a clinical point of view. Therefore, applications in which the long-term stress will be lower than 30 MPa are possible. On the other hand, it must be remembered that the creep rate is time-varying and is highest initially in the first load step and then decreases significantly. Therefore, the process of stress relaxation should also be taken into account—that is, the stress decreases with time while the dimensions remain unchanged. Stress relaxation is a property of polymers. Stress relaxation represents how polymeric materials reduce stress under constant strain. Viscoelastic materials behave in a nonlinear, non-Hookean manner. This nonlinearity is indicated by stress relaxation [42]. This may lead to the assumption that, under clinical conditions, the stress in a retainer made with the tested technique will decrease after a certain period of use.

## 5. Conclusions

The modulus of elasticity is an important parameter for retainers, especially in the initial phases of retention. Another important measured parameter was the bending strength. For this quantity, the ranking was similar to the elastic modulus. The bending strength of 1 mm thick samples was more than twice lower than the bending strength of 1.2 mm thick samples. Moreover, it was several times lower than the bending strength of 0.8 mm thick samples. This suggests that both thicker 1.2 mm and thinner 0.8 mm retainers might have better bending resistance than 1.0 mm retainers which was surprising.

The mechanical properties of 3D printed retainers show that they can be an alternative to metal retainers, and the procedure of making new retainers is very simple and can be carried out in a short time due to existing digital models. They are also cheap and fast to reproduce in case of breaking.

Technological development of materials in this field may replace traditional methods in the future, especially when patients have aesthetic requirements or allergies to metals.

## Figures and Tables

**Figure 1 ijerph-19-05775-f001:**
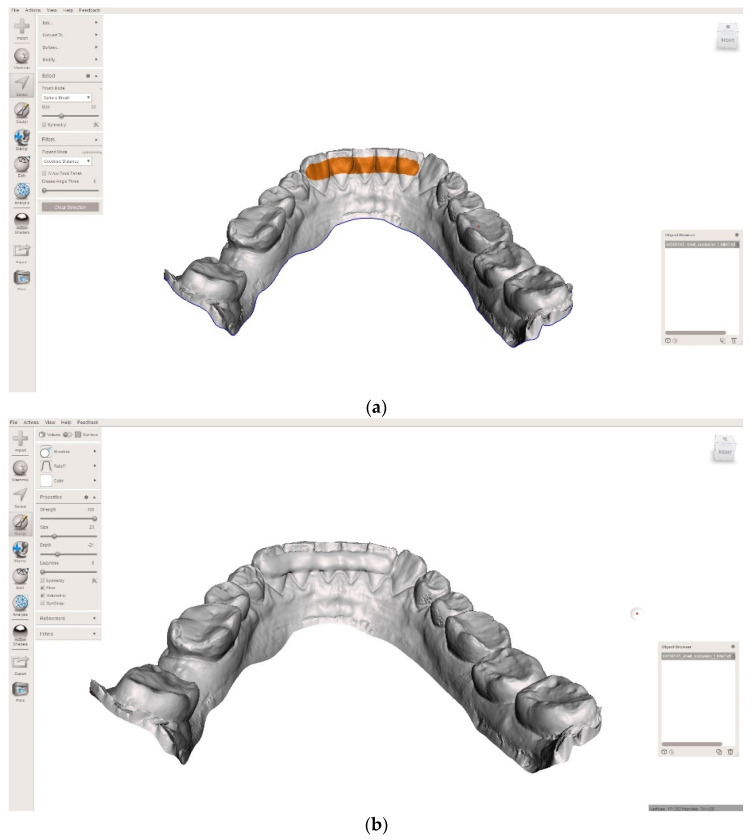
Selected models prepared in Meshmixer software (Autodesk, San Rafael, CA, USA,) to support the production of an individually cemented retainer (**a**) Size of retainer; (**b**) Completed solid model with a retainer of appropriate thickness.

**Figure 2 ijerph-19-05775-f002:**
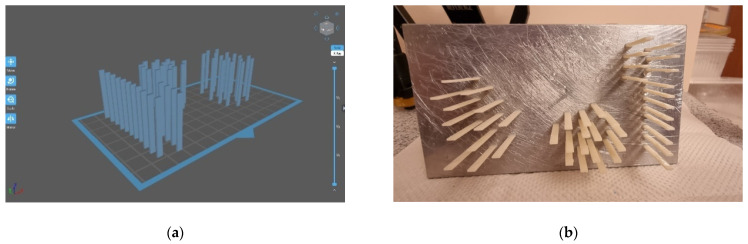
(**a**) Placing prints on building plate; (**b**) First print of 3D—printed blocks on building plate.

**Figure 3 ijerph-19-05775-f003:**
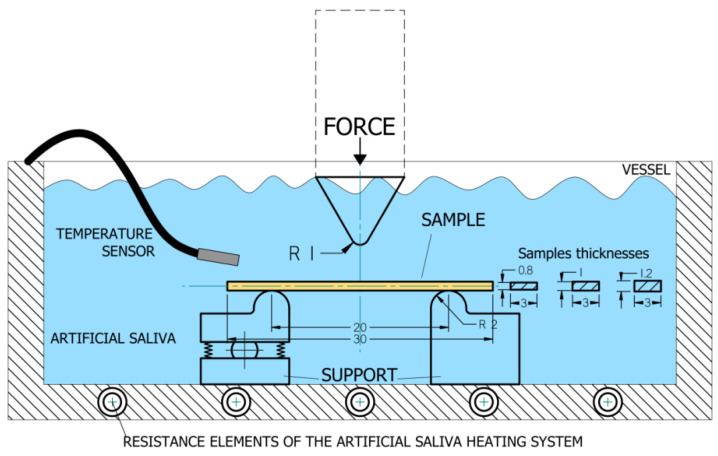
Three-point bending test scheme.

**Figure 4 ijerph-19-05775-f004:**
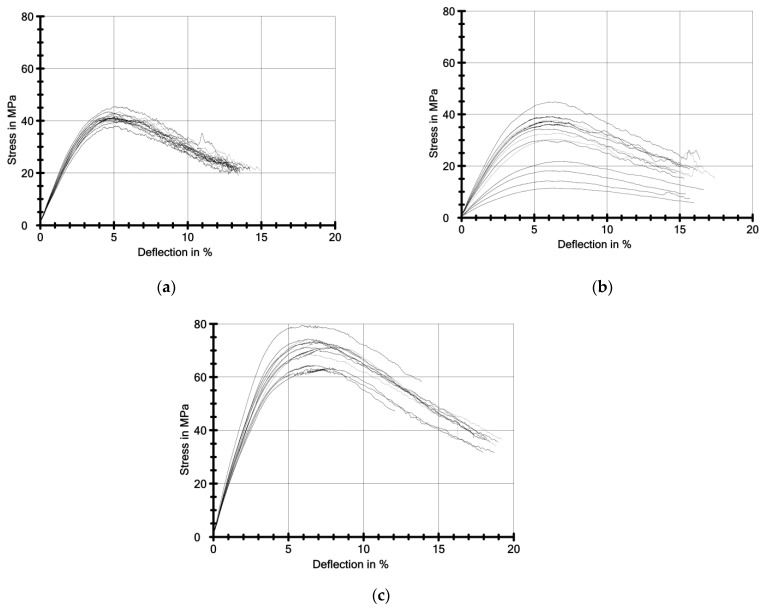
Graphs from the three-point bending test. Nominal samples thickness 0.8 mm (**a**), 1 mm (**b**), and 1.2 mm (**c**).

**Figure 5 ijerph-19-05775-f005:**
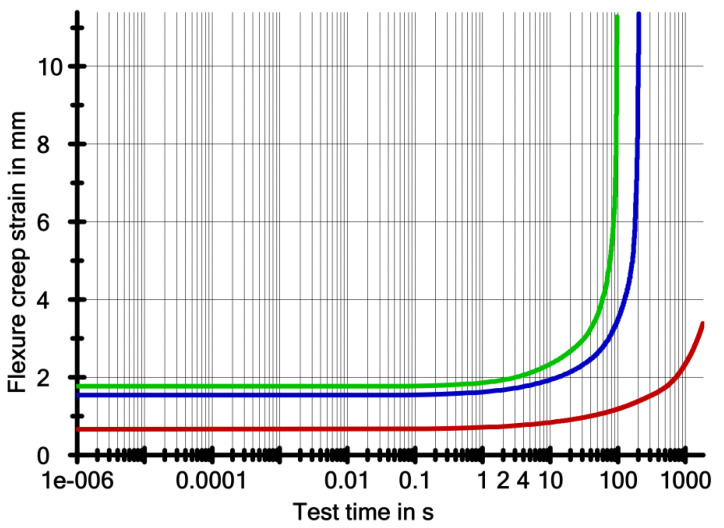
The course of deflection changes over time (logarithmic) during creep.

**Figure 6 ijerph-19-05775-f006:**
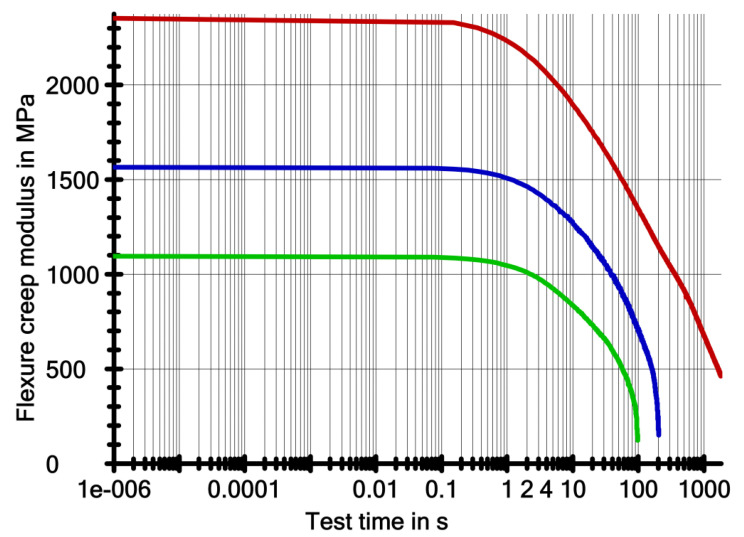
The course of creep modulus changes over time (logarithmic) during creep.

**Table 1 ijerph-19-05775-t001:** K with post-curing Anycubic Wash and Cure.

Material	NextDent C&B MFH
Color	N1
Rinsing in Isopropyl alcohol (min)	4.5
Post-curing (min)	30

**Table 2 ijerph-19-05775-t002:** Printing parameters (Phrozen Tech, Hsinchu City, Taiwan) with resin Nextdent C and B N1 (Vertex-Dental B.V., Soesterber, The Netherlands).

Material	NextDent C&B MFH
Layer Height	0.050 mm
Bottom Layer Count	5
Exposure Time	4.6 s
Transition Layers	6
Transition Type	Linear
Bottom Lift Distance	6 mm
Lifting Distance	6 mm
Lift Speed	60 mm/min
Retract Speed	150 mm/min

**Table 3 ijerph-19-05775-t003:** Descriptive statistics of the results from the three-point bending test.

Parameter	*E_f_*	*σ* _0.2_	*σ_fY_*	*ε_fY_*	*σ_fM_*	*ε_fM_*	*σ_fB_*	*ε_fB_*	*W_fM_*	*W_fB_*
Unit	MPa	MPa	MPa	%	MPa	%	MPa	%	Nmm	Nmm
Samples thickness of 0.8 mm										
$\bar{x}$	1200	34.7	41.7	5.1	41.7	5.1	20.8	13.5	8.45	24.51
s	85.1	1.77	1.91	0.3	1.91	0.3	0.956	0.6	1.07	2.34
ν	7.07	5.11	4.59	6.74	4.59	6.74	4.59	4.52	12.72	9.56
Samples thickness of 1 mm										
$\bar{x}$	786	22.7	29.9	6.2	29.9	6.2	14.9	15.9	9.11	25.55
s	313	7.11	10.4	0.4	10.4	0.4	5.18	0.7	3.11	8.40
ν	39.83	31.32	34.66	6.16	34.66	6.16	34.67	4.66	34.15	32.88
Samples thickness of 1.2 mm										
$\bar{x}$	1950	46.0	70.8	6.9	70.8	6.9	40.4	16.7	29.07	75.98
s	176	4.54	4.75	0.7	4.75	0.7	10.7	3.0	4.40	12.33
ν	9.03	9.86	6.71	9.85	6.71	9.85	26.59	17.64	15.12	16.23

**Table 4 ijerph-19-05775-t004:** Creep test results.

No.	Stage Number	Load Time	*E_t_*	*ε_t_*	*σ_t_*	*σ_fract_*	*ε_fract_*	* _τfract_ *
		min	N/mm^2^	%	MPa	MPa	mm	s
Sample thickness of 0.8 mm								
1	1	1	881.61	3.36	30	20.7	11.4	203.2
	2	3	402.33	7.31	30	-	-	-
	3	6	-	-	-	-	-	-
	4	12	-	-	-	-	-	-
	5	30	-	-	-	-	-	-
Sample thickness of 1 mm								
2	1	1	481.93	6.20	30	20.9	11.3	97.2
	2	3	-	-	-	-	-	-
	3	6	-	-	-	-	-	-
	4	12	-	-	-	-	-	-
	5	30	-	-	-	-	-	-
Sample thickness of 1.2 mm								
3	1	1	1483.77	2.02	30	-	-	-
	2	3	1180.28	2.54	30	-	-	-
	3	6	1001.71	2.99	30	-	-	-
	4	12	796.44	3.76	30	-	-	-
	5	30	466.35	6.38	30	-	-	-

## Data Availability

Not applicable.

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
