# Peer review of "Mechanical Properties of 3D Printed Orthodontic Retainers"

_ijerph, 2022, doi:10.3390/ijerph19095775_

Round 1

Reviewer 1 Report

-Why no null hypotheses are present in the Introduction section?
-In line 207, what does the sentence "Error! No sequence specified" stand for?
-The authors should insert information about the materials and devices that they used (i.e., manufacturer, city, country).
-What is the rationale for using the "Nextdent MFH C&B N1 resin"? Are some papers or chemo-mechanical properties that lead you to that choice?
-Was each experimentation made in the same environmental condition of temperature, relative humidity, and air pressure?
-Please, avoid using vertical rows in tables (e.g., Table A)
-Was the sample size chosen according to previous studies or by a convenience criterion? Was a power analysis run to choose the sample size?
-Please, review the "References" list because some errors are present (e.g., "25. Kr??likowski").
-The authors have to insert a more detailed paragraph in the Discussion section about the limits of their research protocol.
-A review of the English language and style is strongly needed.

Author Response

Thank you very much for your time, careful analysis of the text and all your comments.

We have tried to improve the text according to your comments and our best knowledge.

We trust that our work has become better.

We have marked the responses to each comment in the text in green

  1. Why no null hypotheses are present in the Introduction section?

Null hypotheses was added in the introduction section in the end.

  1. In line 207, what does the sentence "Error! No sequence specified" stand for?

Line was corrected. It is physical formula for creep modulus.

  1. The authors should insert information about the materials and devices that they used (i.e., manufacturer, city, country).

Manufacturer, city, country was added

  1. What is the rationale for using the "Nextdent MFH C&B N1 resin"? Are some papers or chemo-mechanical properties that lead you to that choice?

That is the one of most popular material used for 3D printing long-term restorations in oral cavity. It has all the requirements for the material to be used in human body. Added sentences with explanations.

  1. Was each experimentation made in the same environmental condition of temperature, relative humidity, and air pressure?
  2. Please, avoid using vertical rows in tables (e.g., Table A)

Table was modified to avoid vertical rows

  1. Was the sample size chosen according to previous studies or by a convenience criterion? Was a power analysis run to choose the sample size?

Sample size was chosen according to the tooth size and to be as close as possible to actual size which will be used in oral cavity. We decided to choose 3mm width to fit incisors and thickness which we think would fit retainers needs. Test confirmed that it was good idea to choose those sizes – it represents well the actual retainer which we think might be the future. We added sentence explaining why this size was chosen.

  1. Please, review the "References" list because some errors are present (e.g., "25. Kr??likowski").

Fixed errors in references.

  1. The authors have to insert a more detailed paragraph in the Discussion section about the limits of their research protocol.

Added limitations of the research protocol in the endo of discussions section.

  1. A review of the English language and style is strongly needed.

English was corrected (yellow color)

Reviewer 2 Report

The manuscript in question discusses the additive manufacture of orthodontic retainers made via vat photopolymerisation. The mechanical behaviour of the 3D printed test samples is discussed. 

The content is very well written and in overall its presentation style is quite good. However a couple points require a bit of attention in order for the manuscript to be considered futher for publication. Detailed comments as follows:

  1. A brief mention of conclusions should be indeed part of the abstract but not in a seperate paragraph. Please unify the two accordingly.
  2. There’s no detailed mention of the resin used, in the materials methods section. The authors should clearly explain why they’ve chosen this specific resin. 
  3. Figure C, should be replaced by a table.  Printscreens are not encouraged.
  4. The graphs in Figure E are not really saying much and it’s best to be removed.
  5. In your results and discussion, you have not discussed how your measured properties compare to the values of resin’s manufacturer.
  6. Finally, your conclusions section reads incredibly short. It has to expanded with the real conclusions that this study has go offer.

Author Response

Thank you very much for your time, careful analysis of the text and all your comments.

We have tried to improve the text according to your comments and our best knowledge.

We trust that our work has become better.

We have marked the responses to each comment in the text in green

The manuscript in question discusses the additive manufacture of orthodontic retainers made via vat photopolymerisation. The mechanical behaviour of the 3D printed test samples is discussed. The content is very well written and in overall its presentation style is quite good. However a couple points require a bit of attention in order for the manuscript to be considered futher for publication. Detailed comments as follows:

  1. A brief mention of conclusions should be indeed part of the abstract but not in a seperate paragraph. Please unify the two accordingly.

Added conclusions to abstract section.

  1. There’s no detailed mention of the resin used, in the materials methods section. The authors should clearly explain why they’ve chosen this specific resin. 

Added explanation why this was best choice of resin and what it is used for.

  1. Figure C, should be replaced by a table. Printscreens are not encouraged.

Replaced figure C to the table.

  1. The graphs in Figure E are not really saying much and it’s best to be removed.

Put explanation of figure E showing results.

  1. In your results and discussion, you have not discussed how your measured properties compare to the values of resin’s manufacturer.

We explained how we measured properties using thee-point bending test. We couldn’t find how resin’s manufacturer measured their values (they don’t share this information).

  1. Finally, your conclusions section reads incredibly short. It has to expanded with the real conclusions that this study has go offer.

Expanded conclusions section.

Reviewer 3 Report

1. Generally speaking, the paper is well-written and is interesting. It is structured and laid out satisfactorily. The figures are satisfactory. The utilised references are fine. However, the language, grammar, punctuation, spelling and sentence structures within the current paper, all must be thoroughly assessed and polished to ensure a succinct and coherent read. This is because there are minor discrepancies in the language, grammar, punctuation, spelling and sentence structures. Please obtain the necessary scientific English language reading and editing assistance, if need be, so that the paper has the potential to be read enjoyably by the international readership.

Author Response

Thank you very much for your time, careful analysis of the text and all your comments.

We have tried to improve the text according to your comments and our best knowledge.

We trust that our work has become better.

Generally speaking, the paper is well-written and is interesting. It is structured and laid out satisfactorily. The figures are satisfactory. The utilised references are fine. However, the language, grammar, punctuation, spelling and sentence structures within the current paper, all must be thoroughly assessed and polished to ensure a succinct and coherent read. This is because there are minor discrepancies in the language, grammar, punctuation, spelling and sentence structures. Please obtain the necessary scientific English language reading and editing assistance, if need be, so that the paper has the potential to be read enjoyably by the international readership.

Thank you. Grammar, and language was corrected – most of the language changes were marked in orange and yellow color.

Round 2

Reviewer 1 Report

I am happy with each change made to the manuscript.